# Utilizing a mixed-methods approach to assess implementation fidelity of a group antenatal care trial in Rwanda

Kalee Singh[1]*, Nathalie Murindahabi[2], Elizabeth Butrick[3], Felix Sayinzoga[4‡], David Nzeyimana[2‡], Sabine Musange[2‡], Dilys Walker[3,5]

1 University of California Berkeley School of Public Health, Berkeley, California, United States of America, 2 University of Rwanda School of Public Health, Kigali, Rwanda, 3 Institute of Global Health Sciences, University of California San Francisco, San Francisco, California, United States of America, 4 Maternal, Child and Community Health Division—Institute of HIV/AIDs, Disease Prevention and Control, Rwanda Biomedical Center, Kigali, Rwanda, 5 Department of Obstetrics, Gynecology and Reproductive Sciences, University of California San Francisco, San Francisco, California, United States of America

☯ These authors contributed equally to this work.
‡ FS, DN and SM also contributed equally to this work.
* kalee@berkeley.edu

## Abstract

### Background

The Preterm Birth Initiative (PTBi)–Rwanda conducted a cluster randomized controlled trial to assess the impact of group antenatal care (group ANC) on preterm birth, using a group ANC approach adapted for the Rwanda setting, and implemented in 18 health centers. Previous research showed high overall fidelity of implementation, but lacked correlation with provider self-assessment and left unanswered questions. This study utilizes a mixed-methods approach to study the fidelity with which the health centers' implementation followed the model specified for group ANC.

### Methods

Implementation fidelity was measured using two tools, repeated Model Fidelity Assessments (MFAs) and Activity Reports (ARs) completed by Master Trainers, who visited each health center between 7 and 13 times (9 on average) to provide monitoring and training over 18 months between 2017 and 2019. Each center's MFA item and overall scores were regressed (linear regression) on the time elapsed since the center's start of implementation. The Activity Report (AR) is an open-ended template to record comments on implementation. For the qualitative analysis, the ARs from the times of each center's highest and lowest MFA score were analyzed using thematic analysis. Coding was conducted via Dedoose, with two coders independently reviewing and coding transcripts, followed by joint consensus coding.

**Data Availability Statement:** All relevant data are within the paper and its Supporting Information files.

**Funding:** This trial is supported by the East Africa Preterm Birth Initiative, a multi-year, multi-country

effort generously funded by the Bill and Melinda Gates Foundation (OPP1107312, https://www. gatesfoundation.org/). NM, EB, FS, DN, SM and DW had some portion of their salary supported by the East Africa Preterm Birth Initiative. The funders had no role in study design, data collection and analysis, decision to publish, or preparation of the manuscript.

**Competing interests:** The authors have declared that no competing interests exist.

## Results

A total of 160 MFA reports were included in the analysis. There was a significant positive association between elapsed time since a health center started implementation and greater implementation fidelity (as measured by MFA scores). In the qualitative AR analysis, Master Trainers identified key areas to improve fidelity of implementation, including: group ANC scheduling, preparing the room for group ANC sessions, provider capacity to co-facilitate group ANC, and facilitator knowledge and skills regarding group ANC content and process. These results reveal that monitoring visits are an important part of acquisition and fidelity of the "soft skills" required to effectively implement group ANC and provide an understanding of the elements that may have impacted fidelity as described by Master Trainers.

## Conclusions

For interventions like Group ANC, where "soft-skills" like group facilitation are important, we recommend continuous monitoring and mentoring throughout program implementation to strengthen these new skills, provide corrective feedback and guard against skills decay. We suggest the use of quantitative tools to provide direct measures of implementation fidelity over time and qualitative tools to gain a more complete understanding of what factors influence implementation fidelity. Identifying areas of implementation requiring additional support and mentoring may ensure effective translation of evidence-based interventions into real-world settings.

## Introduction

Successful introduction of any new outpatient care strategy that disrupts the status quo to clinic flow and systems, poses numerous challenges, particularly in low-resource settings. One challenge is to ensure "implementation fidelity," which is defined as the degree to which an intervention is implemented as intended [1]. Implementation fidelity is affected by several factors, including the level of complexity of the intervention. Intervention components impacting complexity include: the number of sessions within an intervention, the number of participants and the incorporation of group level interventions [2–5]. A further threat to implementation fidelity is the decay of skills that is commonly seen after trainings [6, 7]. Considering the many components involved in effectively carrying out an intervention, maintaining implementation fidelity is critical to successfully translate evidence-based interventions into practice [6–8].

The Preterm Birth Initiative (PTBi)–Rwanda conducted a cluster randomized controlled trial of 36 health centers, using a standardized tool to assess number of providers, ANC volume, suitable space for group care, services, and equipment [10]. Eighteen health centers were randomized to receive group ANC while 18 health centers continued to provide individual ANC. The trial assessed the impact of group antenatal care (group ANC) on gestational age at birth, finding no impact [9]. The program previously conducted a process analysis of implementation fidelity [10]. The study analyzed and compared quantitative data from observer completed fidelity monitoring tools and provider self-assessments but found, that while there was overall high model fidelity as assessed by observers, there was poor correlation between observers and provider self-assessment tools. The study recommended future implementation of group ANC/PNC in Rwanda continue collection of self-assessment data, assessment by expert observers and expert coaching and mentoring.

This study analyzes quantitative data from Master Trainer assessments and explores additional information and insights from qualitative reports completed by Master Trainers. The study's aims are two-fold. First, we measure the association between implementation time (experience), and implementation fidelity. Second, we aim to assess whether observer assessment and support contribute to facilitator skills improvement over time. We use our results to inform recommendations for monitoring of group ANC in similar contexts.

The Rwanda model for group care was aligned with the WHO four visit focused ANC visit model [11]. As the model was developed before the WHO recommendation to switch to 8 contacts during pregnancy was made, it was not applied to this program. Visits were spaced eight weeks apart, allowing women to have a predictable group visit schedule. Taking into account health care providers' other responsibilities, the discussion and activity portion of group sessions were limited to 60 minutes [12].

The Rwanda group ANC model was adapted and developed by a local technical working group and global group care experts. The technical working group was comprised of 10 Rwandan maternal-child health stakeholders who met 3 times over 3 months, for 4 to 8 hours each time. The group considered existing evidence around group ANC as well as constraints of their ANC delivery system. Using these data, the group agreed upon priorities, content and structure of the adapted group ANC model [12]. The Rwandan group ANC model was designed to include all the essential elements of the original CenteringPregnancy model [12]. The CenteringPregnancy program, widely considered to be the seminal group care model includes essential elements that ensure facilitative leadership and group processes that encourage participation [12]. One unique element of the Rwanda model is that nurses and CHWs serve as co-facilitators. Nurses and CHWs were trained together in order to reinforce the egalitarian nature of the model. Table 1 below outlines key components of the Rwanda group ANC model.

## Methods

### Ethical statement

Ethical approval for all study activities, including the administration of the Model Fidelity Assessment, was granted by the Rwanda National Ethics Committee (0034/RNEC/2017) and

**Table 1. Model fidelity items.**

| |
|---|
| Demonstrated mastery (accurate knowledge) of the curriculum, including discussion topics and key messages (MFA tool: curriculum knowledge) |
| Followed the lead of the women and could flexibly adjust the visit agenda to better meet women's needs and interests (MFA tool: responsive) |
| Reinforced individual and group accomplishments (MFA tool: praised group) |
| Prepared the group care room environment, including assessment equipment, learning materials, participant refreshment, and indicated medications (MFA tool: room setup) |
| Performed assessments correctly and followed up on abnormal findings (MFA tool: correct assessments) |
| Communicated using language well understood by all participants, and responded appropriately to verbal and non-verbal cues (MFA tool: facilitator communication) |
| Provided ANC/PNC screening, medications and referrals as indicated, consistent with the Rwanda FANC and PNC packages (MFA tool: proper screening) |
| Encouraged active participation in group activities/discussions and paid particular attention to participants who presented as reserved (MFA tool: encouraged participation) |
| Kept time (MFA tool: kept time) |
| Asked open-ended questions to promote discussion (MFA tool: promoted discussion) |
| Ensured that participants spoke more than the co-facilitators spoke (MFA tool: group participation) |

University of California, San Francisco Institutional Review Board (16–21177). A written informed consent form was obtained from each provider and CHW prior to the first group ANC or PNC visit in which she/he participated as a facilitator, for being observed by Master Trainers while facilitating a group ANC or PNC visit. No personal identifiers of providers or CHWs were recorded. Study staff protected all data as confidential. This study analyzed secondary data collected for the purposes of program monitoring.

## Inclusivity in global research

Additional information regarding the ethical, cultural, and scientific considerations specific to inclusivity in global research is included in the S1 Checklist.

## Data collection process

Implementation of the Rwanda group ANC model took place from July of 2017 to May of 2019. A total of 7 Master Trainers, comprised of one nurse, five midwives, and one physician, were trained in group care facilitation, and collaborated in the development of monitoring tools and processes. Master Trainers trained 72 nurses and midwives and 216 CHWs for three days per training cohort on group care facilitation [13]. Master Trainers were scheduled to visit each of the 18 health centers 1, 2, 3, 5, 7, 9, 12, 15, and 18 months after group ANC implementation began.

A concurrent mixed methods design was used. At each visit, the Master Trainer completed two monitoring tools, a quantitative assessment tool called a Model Fidelity Assessment, and a more qualitative report called the Activity Report. Findings from both tools were triangulated with the objective of achieving a more in depth understanding of the relationship between time, observer assessment and support, and implementation fidelity. This methodology is aligned with Creswell's approach to concurrent mixed methods, in which both qualitative and quantitative data are collected during the same stage and are triangulated to more accurately define relationships among variables being studied [14].

The Model Fidelity Assessment was developed prior to initiation of monitoring visits, but the Activity Report was a tool developed by the Master Trainers as monitoring began because they felt a need to give a more holistic assessment. Occasionally separate MFAs were completed on the same day, evaluating different group ANC meetings.

The assessment tool (the MFA) was developed by the Technical Working Group, UCSF's group ANC technical advisor and Rwandan Master Trainers. Items were based on elements from the CenteringPregnancy model that were found to be essential in providing a structure for effective group ANC. The MFA included 15 items: Items one through three recorded the date, observation site and provider code and were not included in the statistical analysis (Table 1). One item (MFA 9), "Husbands and next-of-kin were engaged and participated in activities (if they were present)", was removed, as it was blank in 93% of MFAs [10]. We analyze the remaining 11 items measuring model fidelity.

Each item was ranked using a 5-point Likert scale from 0 to 4. An overall score, ranging from 0 to 4, was calculated by averaging the score of all MFA items. Higher scores assumed the group session was implemented with greater fidelity. The breakdown of scores was: were not able to perform this skill even though the opportunity was present (0), made attempts but need significant help and to be retrained (1), have beginning skills but require more modelling, role-playing, and instruction (2), require a few minor suggestions from the Master Trainer (3), were fully competent (4).

Qualitative data from the Activity Report included responses to open-ended questions on the group care process, lessons learned, best practices, challenges and recommendations.

Upon completion of each session, Master Trainers provided group ANC facilitators with mentoring and support based on monitoring results. Monitoring consisted of observing preparation and execution of the session, after which Master Trainers answered questions and provided feedback focused on bridging gaps. When gaps were observed, Master Trainers conducted repeat visits earlier to observe if the gap had been corrected.

### Statistical analysis of model fidelity assessments

Descriptive statistics describe the characteristics of providers of group ANC, health centers, MFAs and MFA items. The intra-cluster correlation coefficient (ICC) was calculated, based on overall MFA score, to assess for clustering at the health center level. We calculated an ICC of 0.34 based on overall MFA score and thus considered it not necessary to account for clustering via a mixed-effects model in this analysis [15]. Bivariate linear and ordinal regression were performed to measure the association between MFA item score and time since implementation. Time was measured in days since the first monitoring and mentoring visit for each health center. Multivariate regression was performed to measure the association between MFA item score and time since implementation, controlling for individual and health center level predictors. Individual level predictors included provider age, education level and years of experience working in ANC/PNC. Health center level predictors included location (urban versus rural, with rural as the reference group) and patient to staff ratio. Predictors were included based on their potential association with implementing with fidelity the model of group ANC. The p value for statistical significance was set at 0.05. Regression results are presented in years for ease of interpretation. Results were comparable between both models. While the outcome variable of MFA score is ordinal, as assumptions were met for linear regression, we present those results for ease of interpretability.

### Qualitative analysis of activity reports

The objective of the qualitative analysis was to explore what factors Master Trainers perceived as influencing implementation fidelity. For each facility, the highest and lowest scoring MFA and corresponding Activity Report was selected for a total of two Activity Reports per health center. This method of sampling was chosen to compare qualitative findings between each health center's lowest and highest levels of fidelity.

Author KS coded All Activity Reports as did the PTBi Data Manager in Rwanda. As part of the data reduction process, only sections pertaining to implementation fidelity were coded [16, 17]. The coders discussed and reached consensus on the reduced Activity Reports to be coded. The lead coder created a codebook based on the reduced Activity Reports. The codebook was discussed with the second coder until consensus was reached on codes and definitions [18]. Structural coding was used to map categories to relevant items from the MFA. Magnitude coding was used to note whether Master Trainers made positive or negative comments, allowing for the identification of which categories acted as facilitators (positive comments) or barriers (negative comments) [19]. Axial coding was used to draw connections between initial categories and identify themes [20, 21]. Both coders compared and discussed codes until reaching consensus [22]. Dedoose software was used to organize and code the data.

## Results

### Quantitative results

A total of 160 MFAs were completed for 18 health centers over the span of approximately 22 months. The number of MFAs completed per health center ranged from 7 to 13, with an average of 9 (±1.68).

**Table 2. Comparison of model fidelity assessment mean scores with improvement in scores (N = 160).**

| MFA Item | Average Score at Initial Observation | Average Score at Final Observation | Overall Average Score | Bivariate Coefficient |
| --- | --- | --- | --- | --- |
| **Room Setup** | 2.28 | 3.58 | 3.13 | .49 |
| **Kept time** | 2.35 | 3.28 | 2.76 | .47 |
| **Curriculum Knowledge** | 2.39 | 3.56 | 3.16 | .55 |
| **Praised Group** | 2.50 | 3.37 | 3.06 | .45 |
| **Proper Screening** | 2.71 | 3.95 | 3.49 | .42 |
| **Facilitator Communication** | 2.72 | 3.63 | 3.38 | .47 |
| **Responsive** | 2.72 | 3.63 | 3.35 | .46 |
| **Encouraged Participation** | 2.78 | 3.37 | 3.21 | .23 |
| **Promoted Discussion** | 2.89 | 3.47 | 3.24 | .21 |
| **Correct Assessments** | 2.94 | 3.53 | 3.43 | .26 |
| **Group Participation** | 3.06 | 3.26 | 3.15 | .18 |

Between one and three nurses/midwives and/or CHWs were involved in facilitating each group ANC session. When more than one provider was involved in a group session, the variables for provider age, education and years of experience were recorded only for the most senior provider. A total of 59 providers were included in this dataset. Providers ranged from 24 to 51 years old. The majority of providers, 77.1%, were nurses. Seventy-five percent of providers attended university. Previous experience working in ANC/PNC ranged from 0 to 29 years, with an average of 5.69 (± 4.75) years. There were five urban and 13 rural health centers. Patient to staff ratio, on the days health centers offered ANC, ranged from 4.82 to 31.58, with a mean of 11.75 (± 7.18).

Table 2 displays the mean and standard deviation for all MFA items. The averages of all but one of the MFA items were between 3.0 and 4.0 (the maximum possible score). MFA 14 (Kept Time) had the lowest average score of 2.76 and displayed the most variance, with a standard deviation of 1.11. Three health centers in particular appeared to pull down the score for time-keeping, with average MFA 14 scores of 1.9, 2.14 and 2.17. Another relatively low-scoring item was MFA 10 (Praised Group), at 3.06, while high-scoring items were MFA 5 (Facilitator Communication), 6 (Correct Assessments), 13 (Responsive), and 15 (Proper Screening). Indeed, MFA 15 (Proper Screening) had the highest average score of 3.49.

Bivariate regression results displayed an inverse relationship between average MFA score at initial observation and improvements in MFA scores. MFAs 4 (Room Setup), 14 (Kept Time), and 12 (Curriculum Knowledge) had some of the lowest average scores at initial observation and highest increases in scores. MFAs 6 (Correct Assessments), 7 (Encouraged Participation), and 11 (Group Participation) had some of the highest average scores at initial observation and lowest increases in scores (Table 2).

Bivariate regression results showed that a twelve month increase in duration of implementation was associated with an average increase of .37 points in overall MFA score. The highest increases in scores were seen in MFA items 4, 5, 12 and 13 with increases of .49, .47, .55 and .47 respectively. The lowest increases in scores were seen in MFA items 7, 8 and 11, with increases of .23, .21 and .18 respectively (Table 2).

When adding in provider characteristics of age, education level and years of experience working in ANC/PNC, time remained significant for all items (S1 Table). Provider level predictors were not found to be significant for any of the other MFA items.

When adding in health center level predictors of location (urban versus rural) and patient to staff ratio, time remained significant for all items. Health center level predictors were only found to be significant for MFA 8. Scores over time for each MFA item are plotted in S1 Fig.

## Qualitative results

Among 160 MFA Assessments, 134 had corresponding Activity Reports. Activity Report findings shed additional light on key factors affecting implementation fidelity. Six themes emerged from the analysis, including group ANC scheduling, logistics of preparing the room for group ANC, provider capacity to co-facilitate group ANC, knowledge regarding specific content areas, facilitation skills and perceptions of women's experiences with the group ANC process. These themes and key recommendations are summarized in Table 3.

## Provider availability

Eligible health centers were required to have more than one ANC provider available on days when ANC is provided [23]. This requirement aimed to ensure the availability of a designated provider to conduct group ANC.

However, in several instances on days when the Master Trainer was visiting, only a single provider was available to deliver both group and individual ANC. As a result, women visiting the health center for individual ANC (who were not enrolled in the trial), participated in group ANC.

*The provider was the same one to attend to ordinary ANC women, do ultrasound and group care that day. Ordinary ANC women were among group care women.* (Health Center 2 – Lowest MFA Score)

**Table 3. Themes, details and recommendations for group ANC.**

| Themes | Details | Recommendations |
|---|---|---|
| Provider Availability | Limited availability of providers resulted in mixed groups of women from group and individual ANC<br>Group ANC delayed to first accommodate women receiving individual ANC<br>Women arrived late or at incorrect appointment times, resulting in large groups of women of different gestational ages<br>Providers worked collaboratively to provide coverage for both individual and group ANC<br>Nurses were in charge of several services, making it challenging to adequately deliver group care.<br>CHWs were sometimes unavailable to co-facilitate, making it difficult to implement group ANC.<br>Untrained staff occasionally facilitated group ANC when trained staff were unavailable or designated to provide other services. | Cultivate relationships conducive to effective staff collaboration.<br>Effective management strategies are required to balance provision of group ANC with other services.<br>Health center management must develop solutions to balance service needs with staff capacity. |
| Room Preparation | Adequate room preparation among some health centers<br>Staff successfully adapted to prepare rooms in absence of adequate resources.<br>Provision of multiple services, high patient volume and unprepared staff, hindered room preparation. | Health center management and staff must review dates for group care and other services in advance, in order to adequately prepare for group ANC. |
| Facilitation Skills and Process | Several facilitators exceled in leading group ANC.<br>Some facilitators created an environment of judgement and blame, greatly hindering the group care process. | Further training and accountability measures are required to ensure facilitators are delivering group care as intended. |
| Facilitator Content Knowledge | Certain content and skill areas were not well understood and required further training | Additional training must be provided on specific topics. |
| Group ANC Process | Group ANC gave women the opportunity to learn through discussion, shared experiences and relationship building.<br>Negative interactions with facilitators were detrimental to the group experience, resulting in high levels of dissatisfaction among women | Continued supervision, training and accountability of facilitators is essential in ensuring a positive and effective group care experience for women |

In some facilities, providers had the capacity and were able to work collaboratively with the head of the health center to designate individual nurses for both group and individual ANC.

*Even the head of the health center was present helping in the management of the activities related to ANC then avail one nurse to help for day ANC.* (Health Center 6 –Highest MFA Score)

Several challenges were present regarding shortages of trained staff. Nurses responsible for co-facilitating group ANC were also in charge of other services, making it extremely challenging to adequately deliver group care.

*According to the nurse, they have to start by providing emergency care before they join the place where GANC care is taking place.* (Health Center 5 –Lowest MFA Score)

In some instances, not only were nurses occupied with providing other services, CHWs were also unavailable to co-facilitate. Such staff shortages made it difficult to implement group ANC and also disrupted continuity of care, as the same provider was not always available to co-facilitate at subsequent sessions.

*She was also assigned to work in maternity and other trained providers were not available to facilitate the group care. The Provider pointed out that it was not possible that the same group be followed by same facilitator from GANC 2 to GPNC due to the problem of providers' availability.* (Health Center 4 –Second Highest MFA Score)

Untrained staff occasionally facilitated group ANC when trained staff were unavailable or designated to provide other services.

*The nurse who was not trained in group care facilitation was the one in charge of group care that day and says she had been conducting group care facilitation in the past with other facilitators.* (Health Center 7 –Lowest MFA Score)

While several nurses, midwives and CHWs were trained to facilitate group ANC, severe staff shortages resulted in a variety of challenges. On many occasions, group care either began late, was not properly implemented or was facilitated by untrained staff. In addition, other services as well as patient experiences were negatively affected.

## Room preparation

The Technical Working Group recommended group visits be conducted in the interior of the health center, where sessions could be conducted privately. Preparation of the group ANC room included providing weight and blood pressure equipment, learning materials, clean drinking water and indicated medications (such as iron tablets, deworming medication or antimalarial medication). Some health centers were well equipped, with rooms adequately prepared with water, learning materials and a semi-private area for individual assessment.

*Didactic materials were prepared: cards about labor signs, danger signs on a newborn and mother as well as care of them. Materials illustrating birth preparation were ready too.* (Health Center 1 –Highest MFA Score)

In some occasions, when group and individual ANC were provided on the same day, there was confusion around which room to use. High patient volume, in addition to staff unaware that group ANC was being conducted, made it difficult to adequately prepare the room and resulted in individual assessments being conducted in a separate room.

*Had not yet prepared room nor had they decided on which room to use for group care as it was an ordinary day for subsequent antenatal visits. I assisted in arrangement of the room but because it was a walk in for many people coming in and going out, the abdominal assessments were done in close by side room.* (Health Center 4 –Third Lowest MFA Score)

Nurses and CHWs were resourceful in adapting to the challenges of missing materials or broken equipment. However, on multiple occasions, in both low and high performing clinics, staff were unaware that group ANC was being conducted or had challenges effectively coordinating group care preparation with other services being offered.

## Facilitation skills and process

Group care facilitators were trained on leading women through semi-structured activities, with the purpose of creating cohesiveness and trust while generating productive discussion [20]. Facilitators were taught to keep the final objectives of the session in mind, while maintaining awareness of their own biases and opinions. Many facilitators exceled in leading all aspects of group care, from teaching women to measure their blood pressure and weight, to promoting discussion.

*The nurse facilitated the introduction of participants and facilitators as well as the MT (Master Trainer) and initiated the measurement of women's blood pressure and weight. While the health assessment was being performed, the CHW reminded women their group rules before women started discussing among themselves about different experiences on pregnancy.* (Health Center 1 –Highest MFA Score)

Group sessions were difficult to continue when providers were unable to create a safe space for women. Some facilitators spoke more than women and did not allow them time to share their thoughts, approached the group with an attitude of judgement and blame and were disorganized and controlling in discussing content. As a result, women felt intimidated and refrained from participating in discussion.

*He went in wrong direction blaming and requesting them to talk about what is wrong with their pregnancy. Women were intimidated and there was a total silence in the room. The nurse was intervening talking more than women and kept being judgmental.* (Health Center 8– Second Lowest MFA Score)

Facilitators displayed competence in leading group sessions, effectively presenting content while creating an environment conducive to discussion, sharing experiences and establishing relationships among women. Challenges arose when facilitator behavior created a negative environment for women, preventing them from learning in a participatory manner.

## Facilitator content knowledge

While nurses and CHWs underwent training on clinical skills, certain topics, such as preeclampsia, were not well understood and required clarification by the Master Trainer.

*During recap of danger signs that were covered previously, I realized that they did not understand what pre-eclampsia or eclampsia (kugagara was). Neither the CHW nor the nurse could explain it. They both said it is when the mother is stiff and all the blood has stopped flowing in the body and the mother is almost dead. I had to intervene with the early and later signs and what could be the complications, in order to make them understand.* (Health Center 2 –Lowest MFA Score)

Nurses and CHWs appeared to be well versed on group ANC content and skills. Selected clinical components require additional guidance, ensuring accurate assessments and clear explanations of topics.

## Group ANC process

A key tenant of group ANC is the group process, through which women develop trust, cohesion and mutual support. Group ANC gave women the opportunity to learn through discussion and sharing personal experiences. Women were also able to gain a clearer understanding of family planning and what to expect during pregnancy.

*Every mother has given a chance to share with others what have learned from Ibaruke neza mubyeyi including friendship, take care of themselves which enable them to deliver term and healthy babies.* (Health Center 5 –Highest MFA Score)

Negative interactions with facilitators were detrimental to the group experience, as women expressed their dissatisfaction and hesitation to return for subsequent sessions.

*After he left women expressed also their worries as they told me that they can't really talk when he is the one around because he is always nervous and shout at them. When asked if they will come back for GPNC, they were clear and told me if he is the one in there they won't bother coming back.* (Health Center 12 –Second Lowest Score)

Master Trainers' perceptions suggested women had largely positive experiences, using group ANC as a forum to learn and clarify misconceptions regarding pregnancy, while creating supportive relationships through activities and sharing personal stories.

## Discussion

Understanding fidelity of implementation throughout program introduction is critical in assessing which components of group care may require additional training and support, allowing implementors direction for course correction. Utilizing monitoring results to provide focused mentorship can potentially improve fidelity and strengthen the internal validity of the study. In this case, we see that the trial was implemented with sufficient fidelity and study results can be attributed to the intervention itself, as opposed to implementation failure.

Our study findings are aligned with the concepts posited in Carroll's framework for implementation fidelity [24]. The framework suggests interventions where key components are identified in advance, may have higher levels of fidelity compared to less structured interventions. However, intervention complexity must also be considered, with more complex interventions risking variation in fidelity in how different components are implemented. The framework also considers implementation monitoring, followed by feedback and training, as factors potentially improving both quality of delivery and implementation fidelity. The authors argue the suggested strategies are particularly crucial in the case of complex interventions.

Group ANC implementation fidelity in low and upper middle income countries has only been measured in four other studies, in Malawi and Tanzania, Rwanda, Nepal, and Mexico [10, 25–28]. At the time of this study, there is an active randomized clinical trial in Malawi measuring the degree of implementation success and associated contextual factors [29]. The Nepal trial included a component measuring the effect of time on process fidelity. The study used observations from a pre-study pilot to assess fidelity and re-train facilitators on topics, mainly regarding peer-discussion. During the intervention period, fidelity data was collected after every group visit and analyzed quarterly. The study found significant improvements over time for women supporting each other and facilitators providing dedicated time to group sessions instead of engaging in other clinic activities. No time effect was found for women sharing and actively engaging in group sessions. Study results are similar to our study, which found only small significant effects for encouraged participation, promoted discussion and group participation.

This study found that group ANC was delivered overall with high fidelity (mean overall MFA score of 3.18 (±.52)). Regression results found a significant positive association between elapsed time since implementation and fidelity for all MFA items. These findings suggest that the "soft skills" required for group care facilitation can be learned, retained and even improved over time, at least with monitoring and support visits in place. The Nepal and Mexico studies also found the need for continuous monitoring and feedback to improve facilitation skills. The Nepal study initially observed highly didactic facilitation. Researchers noted many nurse-midwives had decades of experience providing one-on-one ANC and were not accustomed to taking into account the social context or patients' beliefs. In addition to the initial two-day training, routine post-session debriefings including real-time feedback from nurse supervisors were required to improve facilitation and ensure success of the group ANC process.

Similarly, the Mexico studies had to adapt provider training by adding additional one-on-one, on-site time to focus on developing facilitative leadership skills. The vertical nature of doctor-patient relationships proved further challenging, with the study finding the item with the lowest level of fidelity to be, "Whether the facilitator introduced her/himself in a friendly non-hierarchical way and guided but did not control the conversation." The study recommended additional training to achieve the participatory approach required for group ANC.

The MFA items with the lowest average scores at initial observation, room setup, curriculum knowledge and kept time, showed the highest improvements over time in the regression analysis. As this trial was a group ANC pilot, we were not surprised by the initial low scores and subsequent room for these items. The MFA items for correct assessment, promoted discussion and group participation, had the highest average scores at initial observation and the least improvements over time in the regression analysis. High initial scores for clinical assessments were expected, due to providers' prior clinical experience. However, high initial scores for facilitation and participation were unexpected due to the innovative nature of group ANC and departure from traditional patient provider interactions. These results seem to indicate that where providers scored lowest initially, they were able to improve their scores and master the necessary skills with practice, while they maintained the skills they performed well on at the onset. We suggest initially high-scoring items may require less frequent mentoring while initially low-scoring items may require more frequent monitoring and mentoring.

Certain items displayed variations in trends over time. The trend line for the average score of keeping time of group ANC sessions showed a steady increase through the first four monitoring and mentoring visits, followed by a steady decrease. This may be due to the fact that over the course of program implementation, the actual number of group ANC visits surpassed the initially planned number of visits, resulting in several logistical challenges for health centers. These results are in line with PTBi-Rwanda's previous study, which found that at least

25% of group care visits were not implemented with fidelity to the intended two hour session, lasting more than two hours [10]. The tool analyzed in the previous study included two distinct open-ended questions for how much time was spent conducting health assessments and on group discussion. These findings may be more useful than MFA results in informing which component of timekeeping needs to be addressed. Comparing results from both studies displays the potential utility of using monitoring tools with different modes of data collection.

A unique component of the Rwanda model of group ANC was co-facilitation by CHWs and providers. The MFA and fidelity tools analyzed in the previous PTBi–Rwanda study included measures of fidelity scores as well as agreement in scores across the two different tools. This study assessed qualitative measures of the group process as well as the effect of time on model fidelity, highlighting the importance of facilitation support and skills-building throughout the implementation process. We recommend further exploration into the CHW-provider interaction as well as other factors potentially impacting facilitation, such as provider age and experience, and potential gender norms and interactions between male facilitators and female participants.

Based on large improvements in scores for MFA items with low scores at initial observation, we recommend future group ANC programs include intensive monitoring and mentoring at program onset, followed by periodic monitoring through the remainder of the program. It is important to note, however, that certain components of the group process may require greater support than others, such as, preparing the room for group ANC, completing the group session within one hour, and strengthening facilitator knowledge of group ANC content. We recommend the use of both quantitative and qualitative monitoring tools to provide a complete assessment of implementation fidelity. While the MFA has several strengths, solely focusing on quantitative results can fail to capture critical elements affecting implementation fidelity.

Carroll's framework identifies participant responsiveness as a potential moderator, with a lack of participant acceptance and/or engagement potentially impacting implementation fidelity. We recommend implementation fidelity monitoring take place at every level of the program, in the context of group ANC, this includes the level of the health center, providers and women. This is particularly important in addressing more challenging components of group care, such as the impact of facilitator gender on active participation from women. We suggest program implementers consider contextually appropriate mechanisms for measuring factors influencing fidelity at the participant level, such as an exit interview or complaints and feedback hotline. In ensuring the involvement of women, these mechanisms serve as a participatory approach to monitoring implementation fidelity while informing recommendations for improving fidelity of the group ANC process.

## Limitations

Seven Master Trainers conducted observations and completed MFAs, however, fidelity of Master Trainer support was not documented, and due to financial and logistical constraints, inter-rater reliability for MFAs was not measured [10]. As a result, the degree of agreement between raters is unknown and it is possible the relationship between time and MFA scores presents a limited view of the results. As Master Trainers provided support primarily for identified gaps, this may explain why items with the lowest fidelity scores at initial visit had the largest increase in fidelity scores over time, however, this may pose limitations in interpreting the relationship between time and initially high scoring items.

A single Master Trainer was assigned to complete Activity Reports and MFAs for their designated health centers for the duration of the program. Due to Master Trainers' workloads and

scheduling challenges, continuity was not always maintained. Master Trainers occasionally completed Activity Reports and MFAs for health centers other than what they were assigned to, potentially contributing to variance in perceptions of fidelity and MFA scores. In addition, due to a small sample size of 160 MFAs, there is an increased likelihood of Type 2 error.

## Conclusion

This study aims to contribute to the limited body of research on monitoring group ANC implementation fidelity, in the context of low and middle income countries. We suggest a greater investment in monitoring and mentoring at program onset, with periodic visits there-after. Our study's findings also display the importance of using both quantitative and qualitative measures to assess implementation fidelity. This is particularly significant in identifying areas of implementation requiring greater support, such as room preparation, keeping time and curriculum knowledge. Future programs must consider contextual factors as well as innovation and flexibility in program monitoring and implementation when designing group ANC programs.

## Supporting information

**S1 Checklist. Inclusivity in global research.**
(DOCX)

**S1 Table. Multivariate regression of overall MFA score and individual MFA items on elapsed time, provider and health center characteristics (N = 160).**
(DOCX)

**S1 Fig. Average MFA score trends across time.**
(DOCX)

## Author Contributions

**Conceptualization:** Kalee Singh, Elizabeth Butrick, Dilys Walker.

**Data curation:** Kalee Singh.

**Formal analysis:** Kalee Singh, Nathalie Murindahabi, Elizabeth Butrick.

**Methodology:** Kalee Singh, Elizabeth Butrick.

**Project administration:** Elizabeth Butrick.

**Writing – original draft:** Kalee Singh.

**Writing – review & editing:** Elizabeth Butrick, Felix Sayinzoga, David Nzeyimana, Sabine Musange, Dilys Walker.

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
