## [Decision Letter · Decision Letter 0]

14 Jun 2023

PONE-D-23-09004Utilizing a Mixed-Methods Approach to Assess Implementation Fidelity of a Group Antenatal Care Trial in RwandaPLOS ONE

Dear Dr. Singh,

Thank you for submitting your manuscript to PLOS ONE. After careful consideration, we feel that it has merit but does not fully meet PLOS ONE’s publication criteria as it currently stands. Therefore, we invite you to submit a revised version of the manuscript that addresses the points raised during the review process.

We look forward to receiving your revised manuscript.

Kind regards,

Hannah Tappis, DrPH, MPH

Academic Editor

PLOS ONE

Journal Requirements:

Reviewers' comments:

Reviewer's Responses to Questions

**Comments to the Author**

1. Is the manuscript technically sound, and do the data support the conclusions?

Reviewer #1: Partly

Reviewer #2: Yes

2. Has the statistical analysis been performed appropriately and rigorously? 

Reviewer #1: I Don't Know

Reviewer #2: I Don't Know

3. Have the authors made all data underlying the findings in their manuscript fully available?

Reviewer #1: Yes

Reviewer #2: Yes

4. Is the manuscript presented in an intelligible fashion and written in standard English?

Reviewer #1: Yes

Reviewer #2: Yes

5. Review Comments to the Author

Reviewer #1: This research examines the fidelity with which a 18 health centres recruited in a trial in Rwanda implemented a group antenatal care intervention and explores whether lapsed time (a proxy for experience) and other factors were associated with improvements in the degree of fidelity. Utilising a quantitative fidelity assessment tool, the research finds that the health centres implemented group ANC with reasonably high fidelity, and that lapsed time (during which group ANC facilitators acquired skills to implement) was significantly associated with improvements in fidelity. There is a dearth of knowledge on fidelity of implementing group ANC, and this paper contributes by reporting implementation of an adapted group ANC model in Rwanda. The paper is well-written and will be of interest to PLoS One readers, particularly in LMICs. It provides useful lessons on using facilitation and coaching for fidelity improvement and applying multi-method tools for monitoring fidelity (applicable to real world settings). However, there are some weaknesses that need addressing before publication – largely related to definition and clarification of issues, and some methodological considerations. Suggested areas for improvement are outlined below.

High-level overarching comments

1. This paper investigates an adapted group ANC model. It is great that the MFA tool for assessing this adapted model was developed by varied actors both local and external, indicating that local relevance considered. However, to provide greater context, the authors should consider giving more details on a) the process of adaptation, and b) the ways in which the adapted model differs from the original. Related to this, kindly conceptualise earlier on, what fidelity to this adapted model should look like (the reader read the MFA items only in the results).

2. The paper cites relevant literature about the intervention (group ANC model) and the few available studies exploring its fidelity. However, since this research is about fidelity (an implementation concept), including some literature on implementation research would strength the context, rationale, and discussion. The discussion could be further strengthened by interpreting the results in the context of implementation frameworks (e.g., the themes from the qualitative AR analysis seems aligned to the moderating factors of implementation fidelity posited in Carrol’s conceptual framework on implementation fidelity. Available literature could also be cited to support the statement in lines 445-447.

3. One of the aims of this study was to assess whether observer assessment and support contribute to facilitator skills improvement over time. The master trainers provided expert coaching and mentoring after each fidelity assessment and AR. It is important for readers to understand the nature of the support provided by master trainers. Please provide this detail. Please also clarify – was the nature of the coaching / mentoring provided documented after each visit, was the support provided consistently after each visit, and was the support provided regardless of observations or was there emphasis on providing support only if tasks scored poorly (was there documentation to show support was provided with fidelity)? It is important to clarify these issues to provide some explanation for some of the results – e.g., the results show facilitation tasks with the lowest fidelity scores at the initial visit had the largest increase in fidelity score over time. Is it possible master trainers emphasised providing coaching only for observed weaknesses? Could this explain why high scoring tasks did not improve over time?

Specific issues for clarification and elaboration

4. Statistical analyses require further elaboration (test statistics used, measures of association applied), and justification for using average IF score over time as the outcome, and whether considered using % change in fidelity score over time.

5. Elaborate the qualitative approach – was this mixed methods or multi-method study. If the latter, kindly clarify type of mixed methods, and how integrated quali and quanti data.

6. Lines 61-63 and 65-67 – suggest three rather than two study aims?

7. Quantitative data showed high levels of average curriculum knowledge. Was this comment about need for additional training, provided early on or later as well?

8. Line 99 – please clarify how the 18 centres were selected, and how did the investigators select the facilitators to assess at the 18 health centres? A line or two to explain the trial within which this study is embedded would be useful.

9. Line 143 – 144 – “…explore what factors influenced implementation fidelity” – consider adding that this was as perceived by the master trainers (since these were their observations).

10. Line 149 – please clarify, did the Ars record observations on other issues besides IF?

11. Line 210 to 231 – the issues reported under the theme ‘group ANC scheduling’ seem to all relate to provider availability / health centre capacity rather than scheduling? Kindly clarify how these issues differ from those reported under the theme ‘provider availability’ on page 14.

12. Lines 123 to 125 - do not mention any section pertaining to IF though?

13. The limitations are well-discussed (though, may want to consider issues raised in point # 3 above as other possible limitations).

Reviewer #2: This manuscript is very timely as there has been inconsistencies in how to measure group care fidelity. The combination of qualitative and quantitative measures makes this a solid approach to understanding the effects of the model. It also includes both perspectives of the facilitators and participants.

6. PLOS authors have the option to publish the peer review history of their article (what does this mean?). If published, this will include your full peer review and any attached files.

Reviewer #1: No

Reviewer #2: No

---

## [Author Response · Author response to Decision Letter 0]

20 Jun 2023

Journal Requirements:

The manuscript has been edited accordingly.

The questionnaire has been uploaded as supporting information.

Supporting information captions have been added to the end of the manuscript.

Seven references were removed as they were part of the broader study, but not this particular paper. One reference (14) was added regarding mixed methods.________________________________________

5. Review Comments to the Author

High-level overarching comments

1. This paper investigates an adapted group ANC model. It is great that the MFA tool for assessing this adapted model was developed by varied actors both local and external, indicating that local relevance considered. However, to provide greater context, the authors should consider giving more details on a) the process of adaptation, and b) the ways in which the adapted model differs from the original. Related to this, kindly conceptualise earlier on, what fidelity to this adapted model should look like (the reader read the MFA items only in the results).

This information has been added in lines 78-82 and Table 1.

2. The paper cites relevant literature about the intervention (group ANC model) and the few available studies exploring its fidelity. However, since this research is about fidelity (an implementation concept), including some literature on implementation research would strength the context, rationale, and discussion. The discussion could be further strengthened by interpreting the results in the context of implementation frameworks (e.g., the themes from the qualitative AR analysis seems aligned to the moderating factors of implementation fidelity posited in Carrol’s conceptual framework on implementation fidelity. Available literature could also be cited to support the statement in lines 445-447.

This information has been added in lines 392-399 and 473-474.

3. One of the aims of this study was to assess whether observer assessment and support contribute to facilitator skills improvement over time. The master trainers provided expert coaching and mentoring after each fidelity assessment and AR. It is important for readers to understand the nature of the support provided by master trainers. Please provide this detail. Please also clarify – was the nature of the coaching / mentoring provided documented after each visit, was the support provided consistently after each visit, and was the support provided regardless of observations or was there emphasis on providing support only if tasks scored poorly (was there documentation to show support was provided with fidelity)? It is important to clarify these issues to provide some explanation for some of the results – e.g., the results show facilitation tasks with the lowest fidelity scores at the initial visit had the largest increase in fidelity score over time. Is it possible master trainers emphasised providing coaching only for observed weaknesses? Could this explain why high scoring tasks did not improve over time? 

Yes, Master Trainers emphasized coaching on observed gaps. There is no official documentation showing that support was provided with fidelity, however, all 7 Masster Trainers were trained in group ANC. This information has been added in lines 147-150 and 487-494.

Specific issues for clarification and elaboration

4. Statistical analyses require further elaboration (test statistics used, measures of association applied), and justification for using average IF score over time as the outcome, and whether considered using % change in fidelity score over time.

Bivariate regression was used to measure the association between each MFA item score and time, while multivariate regression was performed to measure the association between MFA item score and time since implementation, controlling for individual and health center level predictors. The p value for statistical significance was set at 0.05, with a t-statistic of 1.96. This information has been added on lines 157-166, with additional results in the supporting information table.

Average MFA score was provided as a descriptive measure. The models below were run for each MFA item (MFA 1, MFA 2, etc.) as the outcome. 

Model0 � lm(MFA1 ~ `Days Since Beginning Implementation`, data = Model_Fidelity_Data) 

Model1 � lm(MFA1 ~ `Days Since Beginning Implementation` + `Provider Age` + `Provider Education Level` + `Provider Years Experience ANC/PNC`, data = Model_Fidelity_Data)

Model2 � lm(MFA1 ~ `Days Since Beginning Implementation` + `Provider Age` + `Provider Education Level` + `Provider Years Experience ANC/PNC` + UrbanRural + StaffANCRatio, data = Model_Fidelity_Data) 

Looking at percentage change over time would require reducing the data to only a few data points (for example, average MFA 1 score at day zero and final day). Using a regression taking into account all data may provide a more rigorous analysis, also allowing us to assess whether there was a significant association with time, which is not possible looking at percentages alone.

5. Elaborate the qualitative approach – was this mixed methods or multi-method study. If the latter, kindly clarify type of mixed methods, and how integrated quali and quanti data.

This information has been added in lines 117-124.

6. Lines 61-63 and 65-67 – suggest three rather than two study aims?

Edited lines 67-69 to reflect two aims.

7. Quantitative data showed high levels of average curriculum knowledge. Was this comment about need for additional training, provided early on or later as well?

The need for additional training was particularly emphasized by providers immediately following initial implementation (low average score at program onset) and continued throughout.

8. Line 99 – please clarify how the 18 centres were selected, and how did the investigators select the facilitators to assess at the 18 health centres? A line or two to explain the trial within which this study is embedded would be useful.

This information was added in lines 52-55.

9. Line 143 – 144 – “…explore what factors influenced implementation fidelity” – consider adding that this was as perceived by the master trainers (since these were their observations).

Edited to: The objective of the qualitative analysis was to explore what factors Master Trainers perceived as influencing implementation fidelity.

10. Line 149 – please clarify, did the Ars record observations on other issues besides IF?

The report includes several topics, most tied to the implementation process. Please see lines 144-145.: Qualitative data from the Activity Report included responses to open-ended questions on the group care process, lessons learned, best practices, challenges and recommendations. 

11. Line 210 to 231 – the issues reported under the theme ‘group ANC scheduling’ seem to all relate to provider availability / health centre capacity rather than scheduling? Kindly clarify how these issues differ from those reported under the theme ‘provider availability’ on page 14.

Given that scheduling challenges as well as broader challenges in implementing group ANC, both stem from issues related to provider availability, both have been combined under the section Provider Availability, beginning on line 238, and have also been combined in the results table, beginning on line 382. 

12. Lines 123 to 125 - do not mention any section pertaining to IF though?

Lines 121-122 state: Linear and ordinal regression were performed to measure the association between time since implementation and MFA item scores. 

MFA item scores are the measure for implementation fidelity. Please clarify if additional information is needed.

13. The limitations are well-discussed (though, may want to consider issues raised in point # 3 above as other possible limitations). This information was added in lines 487-494.

---

## [Editor Report · Decision Letter 1]

10 Jul 2023

Utilizing a Mixed-Methods Approach to Assess Implementation Fidelity of a Group Antenatal Care Trial in Rwanda

PONE-D-23-09004R1

Dear Dr. Singh,

We’re pleased to inform you that your manuscript has been judged scientifically suitable for publication and will be formally accepted for publication once it meets all outstanding technical requirements.

Kind regards,

Hannah Tappis, DrPH, MPH

Academic Editor

PLOS ONE
---

## [Editor Report · Acceptance letter]

13 Jul 2023

PONE-D-23-09004R1 

Utilizing a Mixed-Methods Approach to Assess Implementation Fidelity of a Group Antenatal Care Trial in Rwanda 

Dear Dr. Singh:

I'm pleased to inform you that your manuscript has been deemed suitable for publication in PLOS ONE. Congratulations! Your manuscript is now with our production department. 

Kind regards, 

on behalf of

Dr. Hannah Tappis 

Academic Editor

PLOS ONE